# Age Estimation of *Dracaena cinnabari* Balf. f. on Socotra Island: A Direct Method to Determine Its Lifespan

**Lucie Bauerová \*, Petr Maděra** **, Martin Šenfeldr and Hana Habrová**

Department of Forest Botany, Dendrology and Geobiocoenology, Faculty of Forestry and Wood Technology, Mendel University in Brno, Zemědělská 1, 613 00 Brno, Czech Republic; petrmad@mendelu.cz (P.M.); martin.senfeldr@mendelu.cz (M.Š.); habrova@mendelu.cz (H.H.)

**\*** Correspondence: lucie.bauerova@mendelu.cz

**Abstract:** Background: *Dracaena cinnabari* is a monocot species that does not form annual tree rings; thus, its age can only be estimated. This species is threatened by low natural regeneration, with an evident absence of younger individuals most likely caused by overgrazing; therefore, knowing trees' ages is important for possible conservation strategies; Methods: Data collection was conducted on the Firmihin Plateau on Socotra Island (Yemen) in 2021, and the diameter at breast height (DBH) of 1077 individuals was measured, the same as those established on monitoring plots 10 years before the current measurement. The 10-year radial stem increment and DBH obtained in 2011 served as a basis for the linear model from which the equations for the age calculation were derived. Results and Conclusions: A direct model of age estimation for *D. cinnabari* was developed. According to the fit model, the age in the first (10.1–15 cm) DBH class was estimated to be 111 years, while that in the last DBH class (90.1–95 cm) was estimated to be 672 years. The results revealed that the previously used indirect methods for *D. cinnabari* age estimation were accurate.

**Keywords:** DBH; dragon tree; radial stem increment; Soqotra; monocotyledonous trees; ontogeny



## 1. Introduction

The *Dracaena* genus is classified under the family Asparagaceae and the subfamily Nolinoideae [1,2]. Of the 190 described *Dracaena* species [3], only 8 (so-called dragon trees) have arborescent forms: apart from *Dracaena cinnabari* Balf. f. (Socotra Island), there are other species: *D. draco* subsp. *draco* L. s.l. (Macaronesia), *D. tamaranae* A. Marrero, R. S. Almeida et M. González-Martín (Gran Canaria), *D. draco* subsp. *ajgal* Benabid et Cuzin (Morocco), *D. draco* subsp. *caboverdeana* Marrero Rodr. and R.S. Almeida (Cape Verde Islands), *D. ombet* Kotschy and Peyr. s.l. (East Africa), *D. serrulata* Baker s.l. (south of the Arabian Peninsula), and *D. schizantha* Baker (Somalia) [4–6]. Most of them are endemic, and the populations they form usually have limited distributions [5,7–9]. Based on the IUCN Red List, many dragon tree species belong to the endangered category [6]. In addition, *Dracaena* species are monocotyledonous plants [4,10]; however, they differ in the present monocot cambium [10], which causes secondary thickening of stems and branches [10,11]. They are also well known for their red resin, which is also known as dragon's blood and was used as a medicine in ancient times [5,6,12].

The species *D. cinnabari* is endemic to Socotra Island (Yemen) [13]. Socotra is an archipelago composed of four islands located in the Indian Ocean and is home to diverse vascular plants; about 37% are endemic [14–16]. Thus, it is not surprising that the Socotra Archipelago was stated a UNESCO Biosphere Reserve in 2003 [17]. The natural character of Socotra Island is the result of long-time isolation from the mainland [18], as well as the geological processes, together with the adaptation of the organisms to extreme weather conditions [19]. Moreover, the grazing and overgrazing of livestock has also likely had some influence [19,20], and it is considered the main cause of the low natural regeneration

of *D. cinnabari* on Socotra Island at this time [16,21,22]. Some authors also attribute it to aridification of the island due to climate change [14,23]. However, we found that *D. cinnabari* exhibited a high percentage of germination of seeds under controlled conditions (e.g., 84.6% at 26 °C), thus showing relatively high potential for natural regeneration [24]. *D. cinnabari* is also an important nurse plant, creating better conditions for other organisms; thus, with its possible decline, other species' numbers would also be reduced [16,25,26]. Despite its importance, its original distribution has been reduced, and according to Attorre et al. [23], it grows in only 5.5% of its potential distribution today. For these reasons, it is crucial to conserve this species.

For effective conservation management and future predictions, it is important to know the ages of this species [27]. Although *Dracaena* species form some tree rings [28], it has not been proven that they correspond to annual growth [28]; thus, they cannot be used to calculate the age of the tree exactly as in temperate tree species and some tropical tree species [29]. Therefore, the ages of dragon trees have always been estimated. The oldest estimates were made for the individual *D. draco* on Tenerife Island in Orotava (with 15 m stem circumference), for which the age was overestimated as thousands of years [30–32]. The largest tree on Tenerife at this time, Drago Milenario in Icod de los Vinos, was estimated to be a maximum of 365 years old [33]. More recent studies [21,27] have estimated age using an indirect method focused on the probability of flowering as a regular phenomenon, based on which the age of the crown is estimated. In other words, flowering gives rise to the growth of new branch sections, and the length of time after flowering before the process repeats indicates the age of a branch section [21,27]; the branch sections are "sausage" shaped and form an "umbrella" crown shape [23,27,34]. This, naturally, means that the age of the crown can then be estimated as the sum of the estimated intervals between each individual flowering event [21]. In *D. cinnabari*, the interval between two flowering events can be from 13.7 to 29.6 years (on average, 18.7 years) [27]. This linear relationship assumes that all branch sections are of the same age, but the earlier method was improved by Adolt et al. [21]. They estimated that the time between the flowering events on the Firmihin Plateau differed with the age of the tree, decreasing from 28 to 10 years between the 1st and 2nd event and between the 25th and 26th event, respectively. However, these are only estimates of crown age. The ages of juvenile trees at the time of first flowering were studied by Maděra et al. [35]. Lengálová et al. [2] used the indirect method of Adolt et al. [21] to estimate the ages of *D. draco* subsp. *caboverdeana* and *D. ombet*. For the former species, the interval between the flowering events was estimated to be 4.9 years, and for the latter species, it was 5.2 years on average.

*D. cinnabari* is not only very old, but also exhibits very slow growth, as mentioned by Maděra et al. [5]. Maděra et al. [13] reported the mean annual height increment of juvenile *D. cinnabari* plants in situ to be only 2.65 cm.

The objective of this study was to calculate the ages of *D. cinnabari* trees on the Firmihin Plateau (Socotra Island) through direct repeated measurement of the radial stem increments as a direct method to determine the age of this species. Until now, only indirect methods to estimate the age of *D. cinnabari* [21,27] have been published; however, the direct method has not yet been applied. The proposed direct method could help to assess the aforementioned indirect methods of age estimation. *D. cinnabari* populations on Socotra Island seem to be mostly mature or overmatured, according to Adolt et al. [21]; thus, knowing the ages of the trees can be very useful for effective conservation. Knowledge of the lifespans of tree species is important for setting time frames for conservation management measures, as noted by Altman et al. [36].

## 2. Materials and Methods

### 2.1. Study Area

Socotra Island is located in the Arabian Sea, between 12°19′ N and 12°42′ N latitude and 53°18′ E and 54°32′ E longitude [15,21,37]. The island has an area of approximately 3600 km$^2$ [15,37]. The arid, tropical climate of the island [14] is influenced by two mon-

soon periods: the southwest/summer monsoon period and northeast/winter monsoon period [38,39]. The summer monsoon begins at approximately the end of April or the beginning of May, when humidity increases to approximately 85%, and it lasts until the end of September or beginning of October [38,39]; however, some precipitation can also fall in this period, sometimes in significant amounts [39]. The winter monsoon starts between the end of October and the first half of November and continues until March, bringing some precipitation [38,39]. The periods between the monsoons are very dry [15].

The measurements were conducted in Firmihin, which is described as a limestone plateau [37] situated in the central–eastern part of Socotra Island [16]. The largest dragon tree forest in the world can be found here [16] (see Figure 1). The *D. cinnabari* forest in Firmihin is also unique to Socotra Island because this species is quite scattered in other areas. Firmihin reaches an altitude of 400 to 760 m a.s.l. [16,20]. The mean annual temperature is 23.4 °C, and the mean annual precipitation is 344 mm [39]. In terms of vegetation, the evergreen shrub *Buxanthus pedicellatus* Tiegh. also grows, forming the *Buxanthus–Dracaena* community; however, *D. cinnabari* predominates [14].

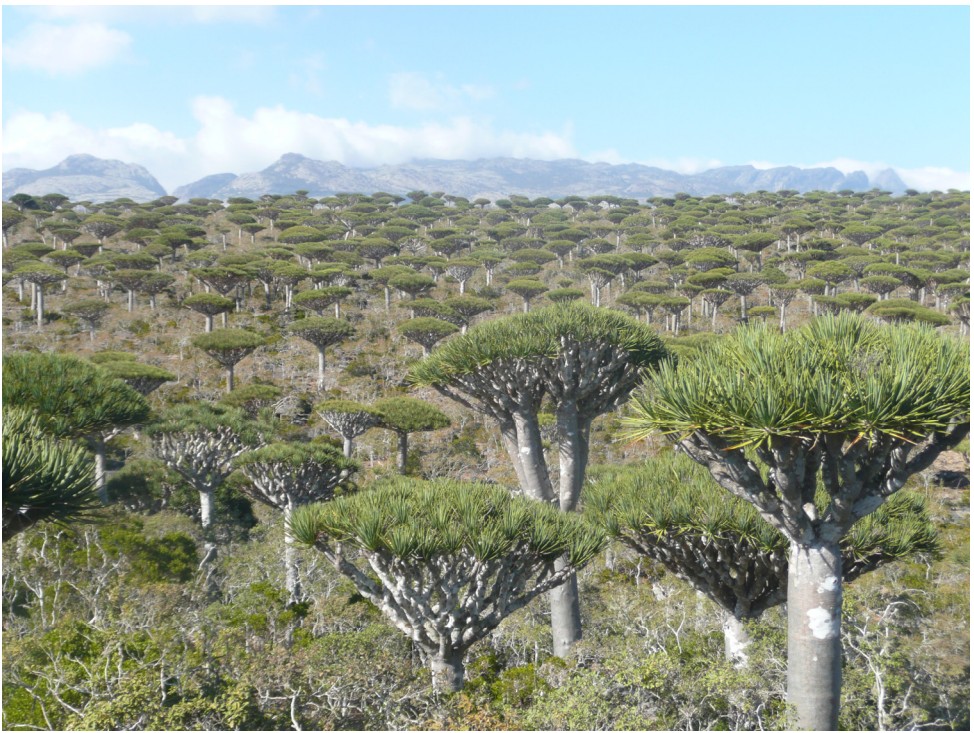

**Figure 1.** Part of the *D. cinnabari* forest on Firmihin Plateau (H. Habrová, 2011).

*2.2. Data Collection*

Adolt et al. [40] conducted an inventory of *D. cinnabari* in Firmihin in 2010 and 2011. These authors covered almost 700 ha of the area of Firmihin, with 107 randomly selected circular monitoring plots with a radius of 25 m. They targeted and measured each of the dragon trees which was higher than 1.3 m in every circular plot. In their study, diameter at breast height (DBH) was measured and branch orders were counted, among the other measured tree characteristics [40]. The branch orders are "sausage" shaped (as mentioned above), and each are separated by a narrowed section; one "sausage" represented one branch order. One branch order represented the period between two flowering events. Adolt et al. [40] used a Field Map device (IFER, CZ) to target the trees. These data served as a basis for our repeated measurements, which were collected after 10 years. We visited the plots from the Adolt et al. [40] forest inventory in March and October 2021 (see Figure 2), and measured the DBH (also at 1.3 m) of the same individuals in individual monitoring plots. The ArcGIS Collector app was used to find the trees and to save the data. The

ArcGIS Collector app helped with navigation by GPS using a mobile phone. The DBH classes (in Results section) were derived from the DBH measured in 2011. Overall, 90 plots were visited, and 1077 trees were measured in total. Seventeen monitoring plots were not measured, for different reasons. Some of them were destroyed by landslides caused by cyclones in 2015 and 2018. In some cases, we were not able to identify the measured trees within the plots, especially in those with high tree density, because the accuracy of the mobile GPS was insufficient to find the exact positions of the plot centers which were needed.

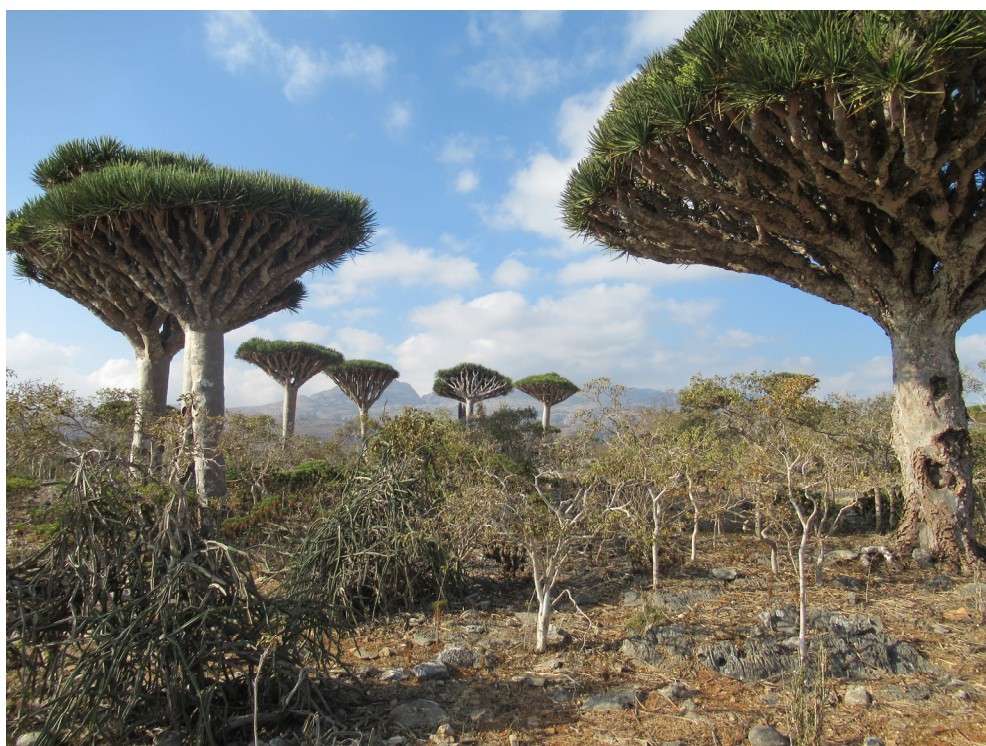

**Figure 2.** One of the monitoring plots in Firmihin (L. Bauerová, 2021).

### 2.3. Age Calculation

We focused on a direct method to determine the age of *D. cinnabari* using the radial stem increment (diameter increment) (RSI) on the same trees at a ten-year interval. The relationship between the 10-year radial stem increment ($RSI_{10}$) and DBH was determined using a simple linear model (Figure 3) with R software [41]. The model was plotted using the ggplot2 R package [42]. The (1)–(3) formulas were derived from the model (Figure 3).

The ($RSI_{10}$) was calculated by the following formula:

$$RSI_{10} = 0.075355 + 0.033060 \times DBH \tag{1}$$

where DBH is diameter at breast height.

The lower 95% confidence interval was calculated by the following formula:

$$RSI_{10(L)} = -0.20367553 + 0.02593465 \times DBH \tag{2}$$

For the upper 95% confidence interval, the following formula was used:

$$RSI_{10(U)} = 0.35438521 + 0.04018616 \times DBH \tag{3}$$

The age ($A_1$) of the trees in the first DBH class (10.1–15 cm) ($DBH_1$) was calculated by the following formula:

$$A_1 = 0.45 \times DBH_1 / (RSI_{10}/10) \tag{4}$$

where 0.45 is the ratio between stem cavity (which is always present in *Dracaena* stems) and DBH (see [35]), and $RSI_{10}/10$ is the mean annual stem radial increment (annual RSI) in the given DBH class.

The ages of the trees in the following DBH classes ($A_{x+1}$) were calculated using the following formula:

$$(A_x) = A_{x-1} + 5/(RSI_{10}/10) \tag{5}$$

where $A_{x-1}$ is the tree age according to the previous DBH class, 5 is the DBH class interval (in cm), and $RSI_{10}/10$ represents the mean annual stem radial increment for the given DBH class (in cm).

The lower 95% confidence interval of age estimation was calculated when $RSI_{10(L)}$ was substituted for $RSI_{10}$ in the formulas used for age estimation. For the calculation of the upper 95% confidence interval of age estimation, $RSI_{10(U)}$ was used.

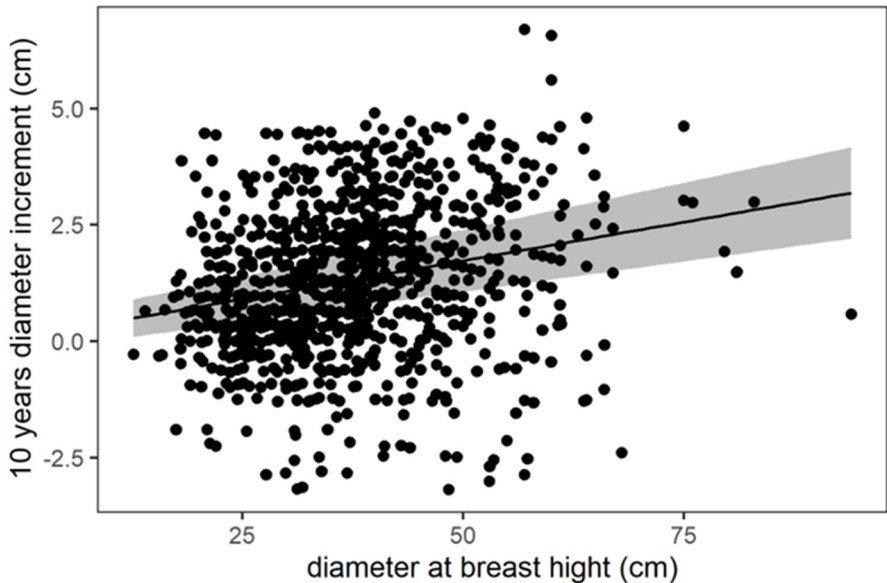

**Figure 3.** Relationship between the 10-year radial stem increment (10-year diameter increment) and DBH (cm), shown by a linear model. The model explains 5% of the data variability.

### 3. Results

#### 3.1. Radial Stem Increment (RSI)

The correlation between the $RSI_{10}$ (cm) and DBH (cm) in 2011 is shown in Figure 3. The linear model (Figure 3) showed a trend of increasing radial stem increment with increasing DBH. The *p* value for the DBH variable was less than 0.0001, less than 0.05; thus, it was statistically significant (Table 1). The $R^2$ was 0.05. The other coefficients are also shown in Table 1. The calculated $RSI_{10}$ (derived from the fitted linear model) in individual DBH classes varied from 0.41 cm (first DBH class, 10.1–15 cm, 95% confidence interval from 0.06 to 0.76 cm) to 3.05 cm (last DBH class 90.1–95 cm, 95% confidence interval from 2.13 cm to 3.97 cm) (Table 2).

**Table 1.** Coefficients derived from the linear model, including the coefficient estimate, standard error, *t* values, and *p* values.

| Coefficients | Coefficient Estimate | Standard Error | *t* Value | *p* Value |
|---|---|---|---|---|
| Intercept | 0.075355 | 0.169486 | 0.445 | 0.657 |
| DBH | 0.033060 | 0.004328 | 7.638 | <0.0001 |

Table 2. The models of $RSI_{10}$ and the ages of the trees with their simulated 95% confidence intervals.

| DBH Class (cm) | Fit Model $RSI_{10}$ (cm) | $RSI_{10}$ Lower Confidence Interval | $RSI_{10}$ Upper Confidence Interval | Age Fit Model | Age Lower Interval | Age Upper Interval |
|---|---|---|---|---|---|---|
| 10.1–15.0 | 0.41 | 0.06 | 0.76 | 111 | 808 | 60 |
| 15.1–20.0 | 0.57 | 0.19 | 0.96 | 198 | 1078 | 112 |
| 20.1–25.0 | 0.74 | 0.32 | 1.16 | 266 | 1237 | 155 |
| 25.1–30.0 | 0.90 | 0.44 | 1.36 | 322 | 1349 | 192 |
| 30.1–35.0 | 1.07 | 0.57 | 1.56 | 369 | 1436 | 224 |
| 35.1–40.0 | 1.23 | 0.70 | 1.76 | 409 | 1507 | 252 |
| 40.1–45.0 | 1.40 | 0.83 | 1.96 | 445 | 1567 | 278 |
| 45.1–50.0 | 1.56 | 0.96 | 2.16 | 477 | 1619 | 301 |
| 50.1–55.0 | 1.73 | 1.09 | 2.36 | 506 | 1665 | 322 |
| 55.1–60.0 | 1.89 | 1.22 | 2.56 | 532 | 1706 | 341 |
| 60.1–65.0 | 2.06 | 1.35 | 2.77 | 557 | 1743 | 359 |
| 65.1–70.0 | 2.22 | 1.48 | 2.97 | 579 | 1777 | 376 |
| 70.1–75.0 | 2.39 | 1.61 | 3.17 | 600 | 1808 | 392 |
| 75.1–80.0 | 2.55 | 1.74 | 3.37 | 619 | 1836 | 407 |
| 80.1–85.0 | 2.72 | 1.87 | 3.57 | 638 | 1863 | 421 |
| 85.1–90.0 | 2.89 | 2.00 | 3.77 | 655 | 1888 | 434 |
| 90.1–95.0 | 3.05 | 2.13 | 3.97 | 672 | 1911 | 447 |

### 3.2. Dracaena cinnabari Age

The age estimates of the *D. cinnabari* trees in different DBH classes according to the fit model, with 95% confidence intervals, are shown in Table 2. The age of the *D. cinnabari* trees, according to the fit model, in the first (10.1–15 cm) DBH class was 111 years. The age of the last (90.1–95 cm) DBH class, according to the fit model, was 672 years.

## 4. Discussion

### 4.1. Radial Stem Increment (RSI)

The annual RSI for the *D. cinnabari* trees varied from 0.041 cm to 0.305 cm. The annual RSI has, thus far, only been measured and published for some *D. draco* species, as also mentioned by Maděra et al. [5]. Pütter [43] and Mägdefrau [33] reported the annual stem diameter increment of *D. draco* to be between 0.2 and 1.13 cm. Other studies were published for 70-year-old (on average) individuals of *D. draco* subsp. *caboverdeana* on the Cape Verde Islands, where the annual RSI varied between 1.05 and 1.25 cm [44]. In addition, Symon [45] reported that *D. draco* in Australia reached an annual stem diameter increment of 1.15 to 2 cm. However, these measurements were obtained from only a few cultivated and, likely, irrigated trees. Nevertheless, *D. draco* probably grows more quickly than *D. cinnabari* [5,13], which may be due to differences in natural conditions (the presence of volcanic bedrock and the more humid climate of the Canary Islands compared to the limestone and more arid climate of Socotra Island) [7].

Regarding direct measurement of the RSI of *D. cinnabari*, there has only been one instance mentioned in the scientific literature thus far [35]. Two plots in Firmihin were repeatedly measured after seven and eight years. The mean annual RSI varied from 0.01 to 0.275 cm in the first plot and from 0.005 to 0.1 cm in the second plot. These values are in agreement with our results.

Maděra et al. [35] mentioned the fact that the annual RSI gradually increases with increasing DBH until reaching the oldest age classes. It was postulated that tree growth must first increase in young and mature trees, and then decrease along with tree size upon reaching an age of senescence (e.g., [46]); however, our results also confirm the unlimited stem radial growth of *Dracaena* (see Figure 3). It seems that *D. cinnabari* first invests more energy in height growth, having, in the juvenile stage, only one leaf rosette [13], and its radial growth is very slow. As it matures, the *Dracaena* tree grows an increasing number of leaf rosettes in the developing crown [21], and, because there is almost no more growth in height, the tree can invest more energy into secondary radial stem growth.

### 4.2. Comparisons of the Dragon Tree Age

Our methodology was focused on calculating the ages of the trees, including both the crown and stem, after reaching at least 1.3 m in height, whereas the aforementioned indirect methods [21,27], based on regular branching, can be used for crown age estimation only. The ages of juvenile trees with only one leaf rosette at the top of the stem can vary from 209 to 549 years; at this point, the vegetative phase ends and the first flowering event occurs. Afterward, thus, crown development begins [35].

The first age estimation for *D. cinnabari* (near Firmihin, at 440 m a.s.l.), by Adolt and Pavliš [27], showed that the crown ages of the oldest trees can be more than 530 years. The improved indirect method by Adolt et al. [21] estimated the oldest crown of *D. cinnabari* in Firmihin to be 500 years, and this was used as a fit confidence interval. In comparison with our data, the oldest tree's age was calculated to be 672 years (a fit 95% confidence interval), including the vegetative phase. Considering that the stem is at least 200 years old [35], both the indirect estimates and our direct age model are highly comparable.

Lengálová et al. [2], using the methodology of Adolt et al. [21], estimated the age of the oldest *D. ombet* crown, with 18 branch orders, to be 94.2 years old in Ethiopia, and that of *D. draco* subsp. *Caboverdeana,* with 22 branch orders, to be 108.6 years old in the Cape Verde Islands. However, as also mentioned by Lengálová et al. [2], the differences between the crown ages of *D. cinnabari* are probably due to the lower number of branch orders within the species. Moreover, *D. ombet* and *D. draco* subsp. *caboverdeana* plots, according to research by Lengálová et al. [2], were at higher altitudes than the Firmihin Plateau on Socotra Island, approximately 1700 m a.s.l. and 1270 m a.s.l., respectively. Additionally, Adolt et al. [21] documented substantial differences in the ages of highland and mountain populations of *D. cinnabari*, with mountain populations growing more quickly and, thus, being younger. The highest crown age was estimated to be 198 years in trees with 30 branch orders, according to the fit model [21]. Therefore, our results are valid only for *D. cinnabari* on the Firmihin Plateau, as at the higher altitudes of Skant, fog is more likely to act as an important source of horizontal precipitation [47], leading to more suitable conditions for growth.

### 4.3. Uncertainties in Age Estimation

Our proposed model does not consider the central cavities in the stems, which act as the rest of the primary tissue until the secondary xylems are formed [28,35,48]. We reduced the first DBH class by 55%, i.e., the mean ratio between the stem cavity and DBH, according to Maděra et al. [35]. This means that the presented direct method overestimated the final ages of *D. cinnabari*. The stems of dragon trees do not begin to increase in radial stem increments from a zero-point, but already have some girth under the forming leaves, as discussed by Maděra et al. [13].

On the other hand, the model did not consider the ages of stems less than 1.3 m above the ground; thus, the model also underestimated the final ages of the trees. Maděra et al. [13] estimated the age of a *D. cinnabari* stem up to 1.3 m tall to be approximately 100 years, in the medium juvenile stage. The late juvenile stage is defined as when the stem reaches more than 1.3 m in height, and this lasts about 100 to 150 years, or more [13]. This stage ends when the first flowering event occurs [13]. However, the authors only guessed the age span of these ontogenetic stages. More precisely, the ages of *D. cinnabari* trees in the juvenile phases were estimated by Maděra et al. [35] to be 209 to 549 years, but the authors mentioned many uncertainties which may have affected this estimation.

Repeated measurement of DBH after 10 years can introduce errors if the measurements are not collected at the exact height on the stems [49] of the investigated trees in both measuring periods. In *D. cinnabari*, repeated DBH measurements can be substantially influenced by wounds occurring in the stem after centuries of resin collection (see Figure 4) by local people [37]. People measuring the DBH tend to avoid these wounds and growth irregularities following wound recovery if they are 1.3 m above the ground. This is probably the reason why we observed greater variability in the results than we expected, including

negative values. Variability within the 95% confidence interval includes that caused by this error in repeated measurement. Removing negative values from our model resulted in an increase in $R^2$ to 0.1, which means that the model explained 10% of the data variability. However, we considered using negative values in our model, because the same DBH measurement error could have been made with positive values.

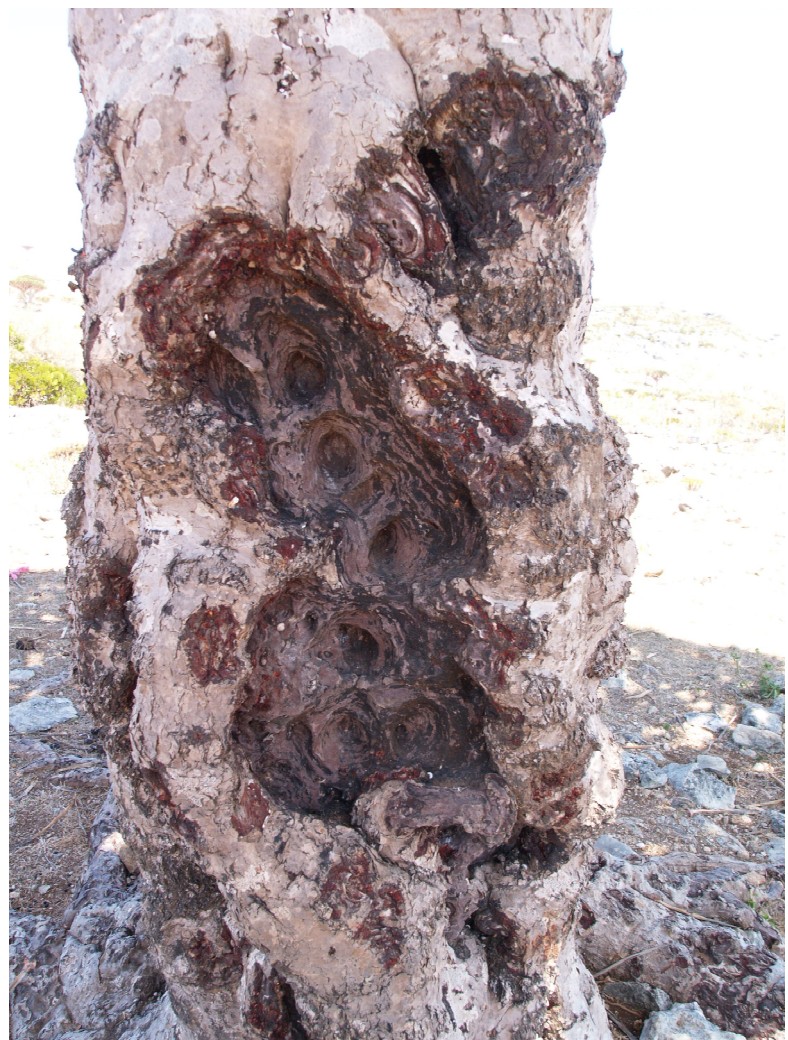

**Figure 4.** Typical wounded stem of a dragon tree, causing mistakes in repeated DBH measurements (P. Maděra, 2021).

Moreover, the individuals were measured in March, which is during the dry season on Socotra Island; therefore, the results could be slightly different in the rainy season when the stems contain more water [50].

Our dataset of repeated measurements of DBH after 10 years showed high variability, and our age calculation model explained only 5% of the data variability. Adding the tree affiliation to the monitoring plots into the model increased the explanation of data variability to 23.6% (see Table S1). This could mean that some environmental factors varying among monitoring plots should be taken into account, because they may have contributed to the variability. Factors derived from DMT (altitude, slope aspect, slope inclination, terrain curvature) included in the model increased the explanation of the data variability by only 3% (Table S1). Thus, there should be other factors, such as soil conditions, that play an important role in dragon tree growth in Firmihin. Soil conditions can affect dragon tree growth very significantly, due to the differences between karst lithosoils and deep clay sediments. Unfortunately, we did not gather such data from the monitoring plots.

Thus, this is a challenge for the next trial which would improve upon this article, which presented the first model of direct age estimation of *D. cinnabari*.

Regardless of the aforementioned large confidence interval caused by the high data variability, our fit model is robust and statistically significant. In the reality, the data variability should be lower, if the measurements of wounded trees, causing both negative and positive mistakes in DBH measurements, would have been excluded. However, we wished to avoid any subjective data manipulation.

Compared with previously published indirect methods of age estimation [21,27], our fit model achieved very similar results. The trend of increasing unlimited radial stem increments with increasing DBH was evident, and the lifespans of the largest trees could reach 672 years (with a 95% confidence interval of 447–1911 years).

## 5. Conclusions

We presented a method of age estimation of *D. cinnabari* trees in Firmihin on Socotra Island to directly determine their lifespans and to verify the indirect methods of age estimation of this species that have already been published. The age estimates obtained using indirect methods did not differ greatly from our fit model for age calculation. This indicates that our direct method confirmed the validity of the indirect methods, and vice versa. Because of the overgrazing and relatively frequent cyclones on Socotra Island in recent years, more conservation measures should be urgently implemented in areas where both natural and artificial regeneration are promoted. Knowledge of the ages of trees helps to determine the time frames for these conservation measures.

**Supplementary Materials:** The following supporting information can be downloaded at: https://www.mdpi.com/article/10.3390/f14040840/s1, Table S1. Different models and their explained variability (represented by $R^2$). Abbreviations: dbhinc—10 years diameter at breast height increment, dbh—diameter at breast height, elv—tree elevation, slo—slope of the plot, asp—slope aspect, shap—plot shape, plot—plot as the factor.

**Author Contributions:** L.B., field data collection, writing—first original draft preparation; P.M., field data collection, writing—original draft preparation; M.Š., data analysis, writing—original draft preparation; H.H., writing—original draft preparation. All authors have read and agreed to the published version of the manuscript..

**Funding:** This research was funded by IGA (Internal Grant Agency) under the Faculty of Forestry and Wood Technology, Mendel University in Brno: project number: IGA-LDF-22-IP-035.

**Data Availability Statement:** All data are available within the article. The original source of the data is saved on the ArcGIS Online private account.

**Acknowledgments:** The research was conducted thanks to the IGA project. We would like to thank the entire team, including the local people of Socotra Island, colleagues from the Czech Republic, and everyone who participated in data collection.

**Conflicts of Interest:** The authors declare no conflict of interest.

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
