# Peer review of "Age Estimation of Dracaena cinnabari Balf. f. on Socotra Island: A Direct Method to Determine Its Lifespan"

_forests, doi:10.3390/f14040840_

Round 1

Reviewer 1 Report

The manuscript is relevant in terms of scientific research and is supported by a clear statistical process, and the results are a significant contribution to science in relation to the species studied.

In the conclusions section it is recommended not to document references, the conclusion should be clarified with the research findings.

Author Response

Thank you

Reviewer 2 Report

It could be interesting that the authors continued to measure Dracaena cinnabarina on Socotra Island to develop a model to determine their age. However, the very poor description of the methods makes it impossible to assess whether the results obtained are valuable. Moreover, the presentation of the results is also questionable. The authors, would have to rework the results of the measurements so it would be possible to assess their scientific value.

Below are detailed comments on the manuscript.

Lines 9-16: Numbers in parentheses, such as (1), (2), (3).... are not needed in the abstract.

L 9: Provide full name of the species: D. cinnabari

L 13: Add the country name for Socotra Island

L 14: This was not “10 years ago” but 10 years before the current measurement

L 16: The manuscript does not contain “the equations for the age calculation”, only the relationship between the radial stem increment and DBH

L 17-18: “DBH class” means range, while in Table 2 there are only selected DBH values, not ranges.

L 32: It is not clear that “dragon tree” is the same that Dracaena cinnabari

L 88-89: Please explain why “knowing the age of the trees can be very useful for effective conservation”

Line 112: What does it mean “between 2010 and 2011”? I think, it should rather be “in 2010 and 2011”

Line 114: Are you sure, it was “diameter of 25 m” or rather radius?

Line 114: They could not measure DBH of all trees but only trees taller than 1.3 m.

Line 116: I think, the word "also" is unnecessary.

Line 116: Explain, what does mean “branch orders were counted”.

Line 119: Please explain exactly how these same trees were found in the field, how you know what DBH they had. The paper by Adolt et al. does not contain such information.

Line 120: Explain how the ArcGIS Collector app was used.

Line 121-122: I do not understand, if “the steep terrain of some plots did not allow the measurements” how in 2011 they measured all trees.

Line 122: I do not understand again, how “seventeen monitoring plots were not found” but you be able to find individual trees on the plot?

L 129: What does mean “(Annual - RSI)”? Why was it placed there?

L 129: It should be explained what RSI10 means.

L 131, 135, 139: It is not explained how formulas 1, 2 and 3 were created.

L 141: What does mean “first DBH class”?

L 153-156: Formula 5 is rather strange and not useful. I think it would be much more useful to provide a general formula for calculating age based on any DBH.

L 168: “5.024e-14” is less even than 0.0001! It is not stated what R2 is.

L 171-172: The numbers given do not agree with those in Table 2.

L 174 (Fig. 1): It is not described what the gray area is. If it is a 95% confidence interval, it should cover 95% of the points. The chart shows that it is much less. It is not clear why so many points (close to half) have a radial stem increment below zero. This is very strange and undermines the reliability of the measurements taken.

L 187 (Table 2): The first column of the table does not contain "DBH class" but specific DBH (every 5 cm). The number 85 is missing. The lower and upper interval for age should probably be swapped. Age has a very large range between the lower and upper 95% intervals (e.g., from 60 to 880 years or from 434 to 1886 years), which shows the questionable usefulness of such an estimate.

L 206-207: Why it is “very interesting”? This is an obvious relationship for trees that the annual RSI increases with increasing DBH.

L 209: “surprisingly”???

Reviewer 3 Report

Dear authors,

I consider that the manuscript is sound as it has scientistic interest considering the analyzed species and the area where it grows. The information provides is relevant, but I consider that the quality of the manuscript should be improve substantially before publication. The introduction mentions information that is not relevant for this work. In the material and methods section one figure showing the sampling places and one picture of the landscape may provide useful information for the readers. The conclusions section should be limited to the conclusions of your research, avoid the use of references and do not mention sentences that do not have any relation with the results. The English should be corrected taking special care in the grammar.

Minor comments to be considered: 

In the abstract, please use the complete scientific name when you mention the species for the first time in the text apart from the title.

In the title, I suggest not to use the author’s initials.

I suggest to find more appropriate keywords that better reflects the content of the manuscript.

It is not convenient to start the introduction with according to…The first sentence should be relevant and consistent.

I do not consider as relevant to mention all the Dracaena species that have arborescent form as an important issue. It is not relevant for this study and does not add relevant information.

Line 38. It is better if you abbreviate the scientific name of the species due to you have mentioned it in the previous paragraph.

Lines 41-44. It is difficult to understand. What is the meaning of character? Of which character are the authors speaking about?

I considered that paragraph starting in line 53 will be more consistent if it starts in line 51.

Lines 56-58. There is a conceptual mistake here as many woody species growing in the tropic also form annual growth rings. For this topic, I recommended:

Worbes, M. 2002. One hundred years of tree-ring research in the tropics – a brief history and an outlook to future challenges. Dendrochronologia 20: 217-231.

Lines 68-71. This sentence has not any meaning as then, you mention the researches that the authors have made.

It is not an appropriate way to start the sentences with the author’s name, more important is their findings, so first of all mention their results and then their names.

The introduction must be checked by a native English speaker as it has many grammar mistakes.

Line 106. In Socotra.

Line 107. The mean annual temperature is..

Line 109. This scientistic name has to have the author’s initials.

Line 115. Do not need to mention that DBH is at 1.3 meters hight as it is globally known.

Line 118. In March and in October 2021 will be better.

Lines 131. and so on.

When you mention a formula, you have to describe each variable you use in it in order to better understand the meaning of the initials. After the formula you have to describe it as follows:

Where: DBH is Diameter at breast height, etc.

Lines 164-165. This sentence belongs to the materials and methods section.

Line 166. You have to mention in the figure the correlation value of this correlation.

Line 181. Please, in the subtitle put the complete name.

The bottom line of table 2 is not complete.

Lines 191-192. You cannot start the discussion with this sentence. It is better if you mentioned your results and then previous studies in comparison with your results.

Line 192. Investigated species. Avoid the use of investigated as it is obvious that you investigate.

Line 199. Do you mean that the other species have better growing conditions? If it is so, you should describe these conditions.

Line 278-282. This sentence does not belong to the conclusions of your results. In the conclusions you do not have to mention previous studies and references.

Reviewer 4 Report

Authors used direct measurements to study changes in DBH before and after 10 years of Dracaena cinnabari in Socotra Island to establish a correlation between DBH and radial growth increment, and thus to predict the age of this dominant species. The results make some scientific sense, but I have the following two questions.

1) The main result of this paper is the regression equation shown in Figure1. Although the regression result passed the statistical test, it was mainly affected by the 7 points with DBH greater than 75 (the rightmost points in the figure 1). If these 7 points were removed, could a robust regression relationship still be obtained? As inferred from other trees with distinct tree-ring characteristics, the dbh-age relationship is not completely linear, I personally suggest that this relationship needs to be carefully considered.

2) L141, the term “DBH class” appears for the first time, indicating that DBH has been classed. Please describe the classification method in detail in the M&M section. Are they classified by 2011 measurements or 2021 measurements?

Overall, I think the regression relationship between DBH and RSI is not particularly robust at present, the subsequent discussion is based on unstable results, I will not comment on it for now. Please confirm the results first.

Round 2

Reviewer 2 Report

The authors included all my comments in their revision. I believe that now the paper meets the requirements of FORESTS and can be accepted.

I have only two minor comments:

(1) Please clarify how the authors obtained the coordinates of the sample plots, which were not included in the paper by Adolt et al.

(2) I suggest further reworking the phrase in line 185: "The p value for the DBH variable is 5.024e-14, which is less than 0.05", I think it could be something like: "The p value for the DBH variable is less than 0.0001". The form "5.024e-14" looks strange for me.

Reviewer 3 Report

This manuscript has improved substantially. The authors addressed all the queries stated about the previous version. Consequently, I suggest that this manuscript has to be accepted.

Author Response

Dear Reviewer,

we thank you for your comments and time.

Reviewer 4 Report

I have no more comments.

Author Response

(The authors gave the same response as above.)
